# Local spectroscopy of a gate-switchable moiré quantum anomalous Hall insulator

Canxun Zhang [1,2,3,6], Tiancong Zhu[1,2,6] ✉, Tomohiro Soejima[1,6], Salman Kahn [1,2,6], Kenji Watanabe [4], Takashi Taniguchi [5], Alex Zettl[1,2,3], Feng Wang [1,2,3], Michael P. Zaletel[1,2] ✉ & Michael F. Crommie [1,2,3] ✉

In recent years, correlated insulating states, unconventional super-conductivity, and topologically non-trivial phases have all been observed in several moiré heterostructures. However, understanding of the physical mechanisms behind these phenomena is hampered by the lack of local electronic structure data. Here, we use scanning tunnelling microscopy and spectroscopy to demonstrate how the interplay between correlation, topology, and local atomic structure determines the behaviour of electron-doped twisted monolayer–bilayer graphene. Through gate- and magnetic field-dependent measurements, we observe local spectroscopic signatures indicating a quantum anomalous Hall insulating state with a total Chern number of ±2 at a doping level of three electrons per moiré unit cell. We show that the sign of the Chern number and associated magnetism can be electrostatically switched only over a limited range of twist angle and sample hetero-strain values. This results from a competition between the orbital magnetization of filled bulk bands and chiral edge states, which is sensitive to strain-induced distortions in the moiré superlattice.

Van der Waals stacking of twisted two-dimensional (2D) atomic sheets provides a versatile platform for engineering exotic electronic states through rotational misalignment that folds dispersive electronic bands into flat mini-bands within a moiré Brillouin zone[1,2]. The resulting suppression of kinetic energy relative to electron–electron interactions can lead to correlated insulating states as well as unconventional superconductivity[3,4]. Moiré flat bands also inherit the large Berry curvature of the individual atomic layers which can result in topologically non-trivial phases[5–7]. Electron-doped twisted monolayer–bilayer graphene (tMBLG)—a graphene monolayer rotationally misaligned with a Bernal-stacked bilayer—stands out among these since it exhibits the quantum anomalous Hall (QAH) effect (i.e., quantized Hall conductance in the absence of external magnetic field) accompanied by doping-controlled switching of its Chern number, an effect not observed in other moiré QAH systems[8]. Such behaviour is expected to be sensitive to local structural parameters such as twist angle and hetero-strain (i.e., the relative strain between adjacent layers). For example, twist angle directly affects the moiré mini-band structure while even small hetero-strains (<0.5%) can be magnified by the moiré superlattice to induce large moiré distortions, thus altering the energetics of mini-bands and the behaviour of emergent correlated and topological phases[9,10]. Understanding the rich physics of moiré systems requires understanding the relationship between exotic electronic phases and local structure, something difficult to achieve using macroscopic probes that only explore spatially-averaged behaviour.

Here we show how scanning tunnelling microscopy and spectroscopy (STM/STS) enables determination of how changes in local structure alter correlated and topological electronic behaviour in

[1]Department of Physics, University of California, Berkeley, CA 94720, USA. [2]Materials Sciences Division, Lawrence Berkeley National Laboratory, Berkeley, CA 94720, USA. [3]Kavli Energy NanoScience Institute at the University of California, Berkeley and the Lawrence Berkeley National Laboratory, Berkeley, CA 94720, USA. [4]Research Center for Electronic and Optical Materials, National Institute for Materials Science, 1-1 Namiki, Tsukuba 305-0044, Japan. [5]Research Center for Materials Nanoarchitectonics, National Institute for Materials Science, 1-1 Namiki, Tsukuba 305-0044, Japan. [6]These authors contributed equally: Canxun Zhang, Tiancong Zhu, Tomohiro Soejima, Salman Kahn. ✉e-mail: tiancongzhu@berkeley.edu; mikezaletel@berkeley.edu; crommie@berkeley.edu

tMBLG field-effect transistor devices. We find that tuning the electron doping concentration of tMBLG results in the emergence of charge gaps observable to STS at filling levels $\nu = 2$ and $\nu = 3$ (i.e., two and three electrons per moiré unit cell), indicating the formation of correlated insulating states. STS performed in an out-of-plane magnetic field allows us to detect non-trivial topology in the $\nu = 3$ QAH insulating state which has total Chern number $C_{tot} = \pm 2$, and to demonstrate its dependence on local twist angle and hetero-strain. In addition to observing strong variation of correlation and topological properties at different twist angles, we find that regions having nearly identical twist angle but different hetero-strain values exhibit very different behaviour. In the small-strain regime, the correlation gap evolves into two separate gaps at different gate voltages that correspond to $C_{tot} = +2$ and $C_{tot} = -2$, indicating doping-controlled switching of valley polarization consistent with previous electrical transport results[8]. Such behaviour is absent, however, when large hetero-strain is present, in which case only a single correlation gap with $C_{tot} = +2$ is observed. This behaviour can be understood using a continuum model for tMBLG that reveals how Chern number switching results from a competition between the bulk and edge contributions to orbital magnetization that is highly sensitive to local hetero-strain. These results demonstrate the crucial role that local structural parameters play in shaping correlation and topological effects in twisted moiré systems.

## Results

### Correlated insulating behaviour at integer fillings

Figure 1a shows a schematic of our experiment, which incorporates a gate-tunable graphene device into an STM measurement geometry.

A Bernal-stacked bilayer graphene is placed on top of a monolayer graphene with a twist angle $\theta$ between them, and the stack is supported by a hexagonal boron nitride (hBN) substrate placed on a Si/SiO$_2$ wafer (Methods, Supplementary Fig. 1). The carrier density $n$ of the graphene stack can be tuned continuously via voltage $V_G$ applied to the Si back-gate. Our devices were annealed in ultra-high vacuum before being loaded into the STM system at $T = 4.7$ K for measurement (Methods). Figure 1b shows a representative topographic image of the monolayer–bilayer moiré pattern which exhibits an average wavelength of $l_M = 11.2$ nm, from which we extracted a local twist angle of $\theta = 1.25°$ (Methods). Within each moiré unit cell (dashed box) we observe three representative regions with different apparent heights that correspond to the three local tMBLG stacking orders: BAB, ABC, and AAB (Supplementary Note 1).

We access correlated electronic states of tMBLG by tuning the carrier concentration via $V_G$ and performing d$I$/d$V$ spectroscopy. Figure 1c shows a density plot of gate-dependent d$I$/d$V$ spectra obtained in the BAB region. Estimation of the device capacitance allows us to convert $V_G$ to the filling factor $\nu$, defined as the average number of electrons/holes per moiré unit cell referenced to charge neutrality (Methods). At $\nu = 0$ ($V_G = 0$ V) we observe two narrow peaks in the d$I$/d$V$ spectrum that are centred at $V_{Bias} = 6$ mV and $V_{Bias} = -14$ mV (Fig. 1d) that we identify as originating from van Hove singularities of the fourfold degenerate conduction flat band (CFB) and valence flat band (VFB). Increasing $V_G$ leads to partial occupation of the CFB and shifts both peaks toward lower energy. As the filling level approaches $\nu = 2$ the CFB peak gradually splits into two branches, CFB− and CFB+, that are located below and above the Fermi energy $E_F$ ($V_{Bias} = 0$ mV). At $\nu = 2$

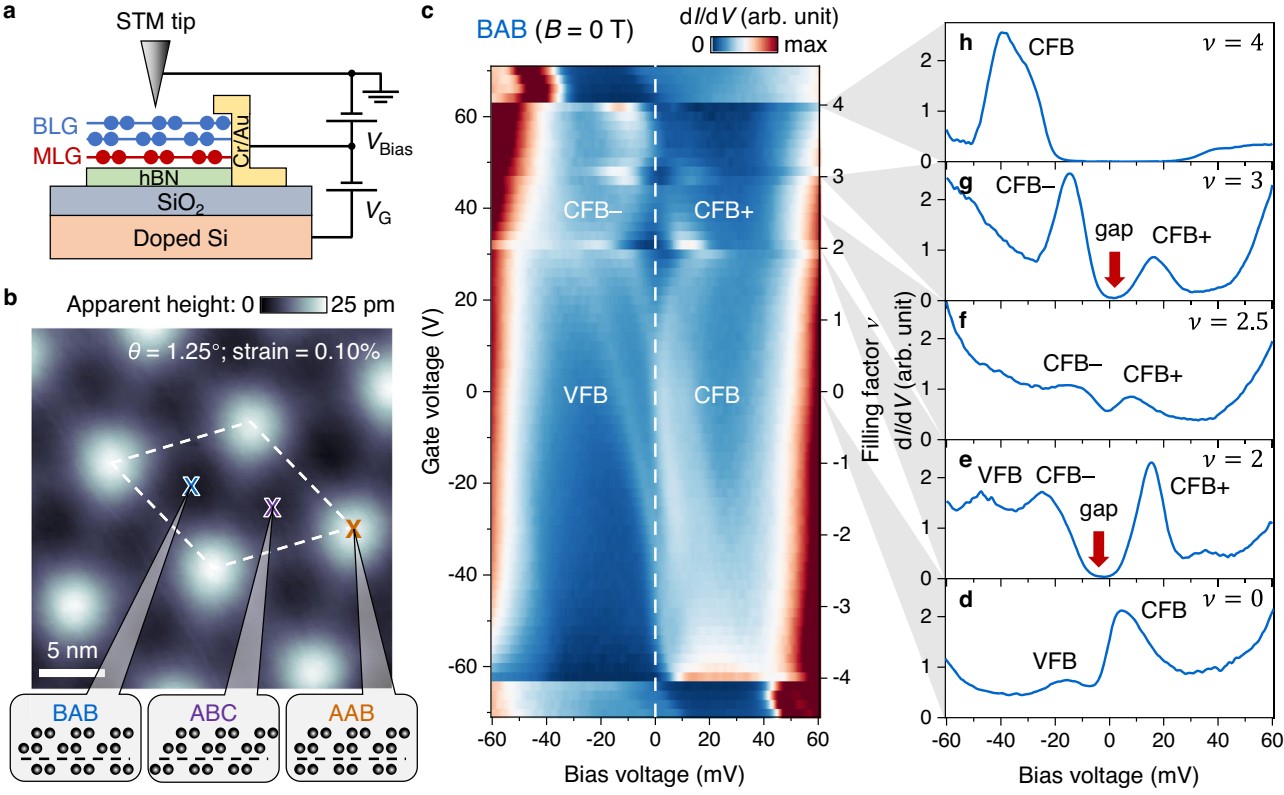

**Fig. 1 | Correlated insulating states in gate-tunable tMBLG. a** Schematic of the gate-tunable tMBLG device used in our STM/STS measurements. MLG monolayer graphene, BLG Bernal-stacked bilayer graphene. $V_{Bias}$ is the sample bias voltage and $V_G$ is the gate voltage referenced to the sample. **b** Representative STM topographic image of tMBLG ($V_{Bias} = -1$ V, tunnelling current $I_0 = 0.02$ nA). The dashed box outlines the moiré unit cell. The local stacking orders BAB, ABC, and AAB are shown in the side view. **c** Gate-dependent d$I$/d$V$ density plot for the BAB stacking region

over the gate range $-70$ V $\leq V_G \leq 70$ V. The vertical dashed line denotes the Fermi energy. **d–h** d$I$/d$V$ spectra measured at **d** $V_G = 0$ V ($\nu = 0$), **e** $V_G = 31.5$ V ($\nu = 2$), **f** $V_G = 39$ V ($\nu = 2.5$), **g** $V_G = 47$ V ($\nu = 3$), and **h** $V_G = 62.5$ V ($\nu = 4$). Spectroscopy parameters: modulation voltage $V_{RMS} = 1$ mV; setpoint $V_{Bias} = 100$ mV, $I_0 = 1.15$ nA for $-70$ V $\leq V_G \leq -2$ V in (**c**); setpoint $V_{Bias} = -100$ mV, $I_0 = 0.8$ nA for $0$ V $\leq V_G \leq 70$ V in (**c**) and (**d**); setpoint $V_{Bias} = -60$ mV, $I_0 = 0.5$ nA for (**e–h**). VFB valence flat band, CFB conduction flat band, CFB− lower branch of CFB, CFB + upper branch of CFB.

($V_G$ = 31.5 V) these two branches have roughly the same spectral weight and a clear charge gap can be observed across $E_F$ (Fig. 1e). As the doping level is further increased from $\nu$ = 2 to $\nu$ = 2.5 ($V_G$ = 39 V) the energy splitting between CFB− and CFB+ becomes smaller and the gap feature evolves into a shallow dip (Fig. 1f). At $\nu$ = 3 ($V_G$ = 47 V) an insulating gap reappears at $E_F$ with CFB− having significantly greater weight compared to CFB+ (Fig. 1g). Finally, at $\nu$ = 4 ($V_G$ = 62.5 V, full filling of the CFB) the CFB− and CFB+ branches merge into a single peak that lies completely below $E_F$ (Fig. 1h). The presence of charge gaps at $\nu$ = 2 and $\nu$ = 3 demonstrates the formation of correlated insulating states at these filling factors, corroborating results from previous electrical transport studies[8,11–13] (Supplementary Note 2, Supplementary Figs. 2, 3).

## Gate-switchable QAH insulating state

To discern the nature of the $\nu$ = 2, 3 correlated insulating states in tMBLG, we applied an out-of-plane magnetic field $\mathbf{B} = (0, 0, B)$ to our sample and performed gate-dependent d$I$/d$V$ spectroscopy. Figure 2a–c shows density plots of gate-dependent d$I$/d$V$ spectra measured near $\nu$ = 2 for $B$ = 0, 1, and 2 T, respectively. The insulating gap feature, marked by vanishing d$I$/d$V$ at $E_F$ and maximum CFB peak splitting, always appears at the same filling level (white arrows) regardless of the $B$ value. d$I$/d$V$ spectra measured near $\nu$ = 3 (Fig. 2d–i), however, exhibit very different field-dependent behaviour. The charge gap (white arrows) is seen to remain constant in energy splitting

(Supplementary Note 3) but to evolve into two separate gaps for $B > 0$ T. These two gaps bracket $\nu$ = 3 and split away from it as $B$ increases.

We can better visualize the magnetic field evolution of the $\nu$ = 3 correlated insulating state by plotting normalized d$I$/d$V$ at $V_{Bias}$ = 0 mV ($E_F$) as a function of both $\nu$ ($V_G$) and $B$ (Fig. 2j). The dark region in the plot indicates vanishing d$I$/d$V$ due to the emergence of a charge gap, which forms a V-shape (white dashed lines) that is roughly symmetric about the $\nu$ = 3 horizontal line. This linear scaling of the correlation gap position with magnetic field is reminiscent of correlated Chern insulating states reported previously in magic-angle twisted bilayer graphene (MA-tBLG)[14–16] where the change in carrier concentration $n$ of a Chern insulating state is related to the out-of-plane field $B$ through the Středa formula $\frac{\Delta n}{\Delta B} = \frac{C_{tot}}{\Phi_0}$ ($C_{tot}$ is the total Chern number and $\Phi_0 = h/e$ is the magnetic flux quantum)[17]. Our observations thus imply that the $\nu$ = 3 insulating state in tMBLG has Chern number $C_{tot} = \pm2$ as derived from the slope of the lines in Fig. 2j. Two significant differences, however, distinguish our results from those reported in MA-tBLG. First, $\Delta n/\Delta B$ linear scaling is observed in MA-tBLG only under high external fields ($B > 3$ T) that break time-reversal symmetry and stabilize the Chern insulating states, whereas such behaviour in tMBLG can be resolved in fields as low as $B$ = 0.2 T (Supplementary Fig. 4) and can be traced back to $B$ = 0 T. Combined with the robust charge gap at $\nu$ = 3, this indicates that the zero-field ground state of tMBLG is a topologically non-trivial QAH insulator with spontaneous time-reversal

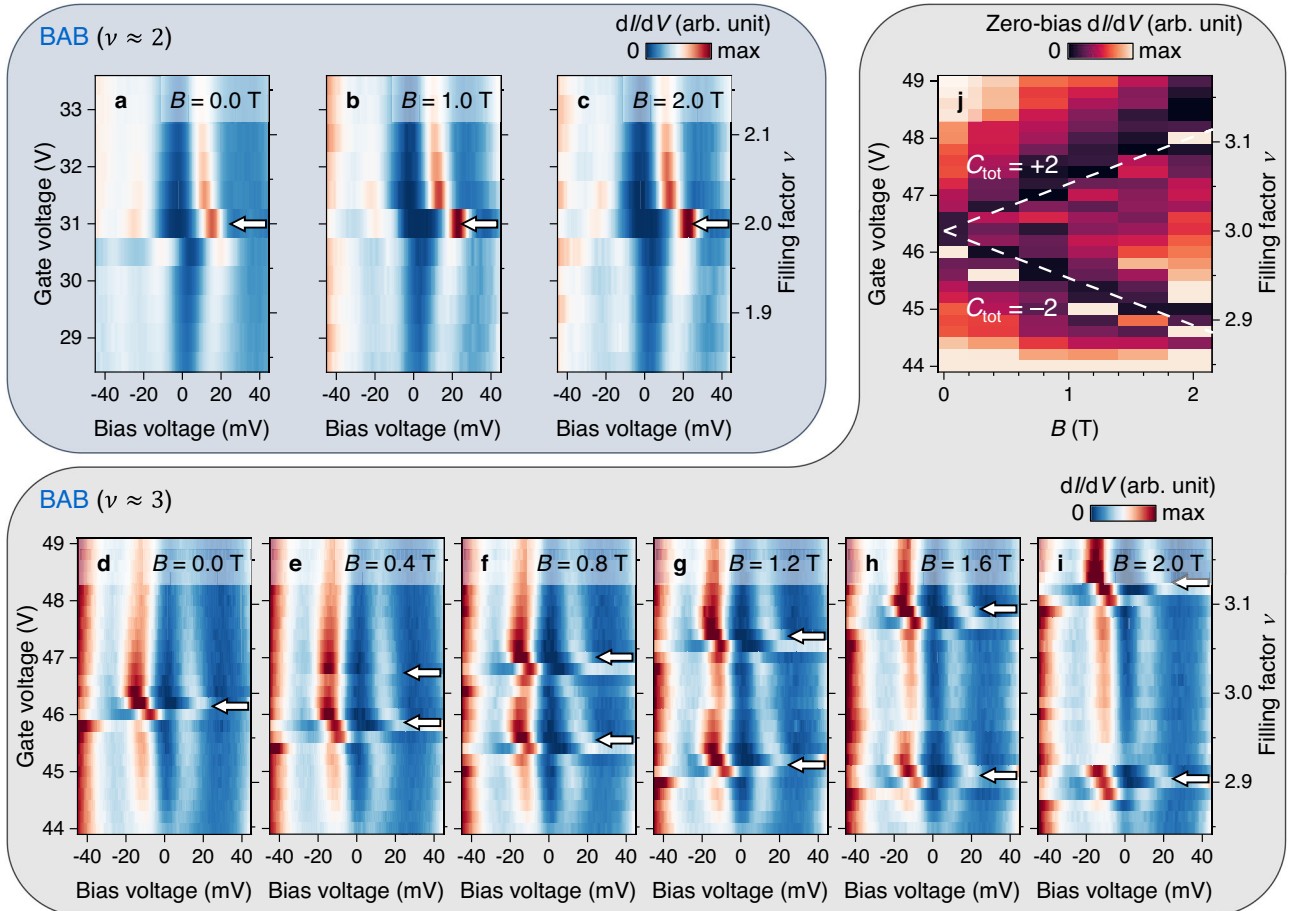

**Fig. 2 | Topological behaviour of correlated insulating states in an out-of-plane magnetic field. a–c** Gate-dependent d$I$/d$V$ density plot for the BAB region near $\nu$ = 2 at **a** $B$ = 0.0 T, **b** $B$ = 1.0 T, and **c** $B$ = 2.0 T (modulation voltage $V_{RMS}$ = 1 mV; setpoint $V_{Bias}$ = −60 mV, $I_0$ = 0.5 nA). Arrows indicate correlation gaps. **d–i** Gate-dependent d$I$/d$V$ density plot for the BAB region near $\nu$ = 3 at **d** $B$ = 0.0 T, **e** $B$ = 0.4 T, **f** $B$ = 0.8 T, **g** $B$ = 1.2 T, **h** $B$ = 1.6 T, and **i** $B$ = 2.0 T (modulation voltage $V_{RMS}$ = 1 mV; setpoint $V_{Bias}$ = −60 mV, $I_0$ = 0.1 nA). Arrows indicate correlation gaps. **j** Normalized d$I$/d$V$ at $V_{Bias}$ = 0 mV ($E_F$) as a function of gate voltage and magnetic field. Dashed lines are guides to the eye following the Středa formula with total Chern number $C_{tot} = \pm2$.

symmetry breaking (Supplementary Note 4). Second, each non-zero integer filling of MA-tBLG features only a single correlation gap that shifts monotonically with increasing $B$, while in tMBLG we observe two separate gaps corresponding to $C_{tot} = +2$ and $C_{tot} = -2$. This indicates that the total Chern number for the tMBLG QAH state is switchable between +2 and −2 by simply tuning the carrier concentration across $\nu = 3$[8].

### Tuning Chern number switching with twist angle and strain

Simultaneous structural measurement via STM and local electronic characterization via STS provide a unique opportunity to investigate how correlation and topological effects in tMBLG are affected by local structural variations at the moiré scale. Figure 3a summarizes our experimental results as a function of local twist angle and local hetero-strain obtained through analysis of moiré anisotropy in our STM topographs (Methods). While the emergence of an insulating gap at $\nu = 2$ is robust in all of our data, the behaviour at $\nu = 3$ depends strongly on both local twist angle and local hetero-strain. Gate tunability of the Chern number is only observed when the twist angle is between 1.25° and 1.28°, as indicated by the green data points in Fig. 3a. When the twist angle increases slightly above this range (orange data points) the charge gap at $\nu = 3$ persists but Chern number switching is suppressed. Figure 3b–d shows representative data from this regime in which the gap feature (white arrows) evolves monotonically toward higher filling

factors as the magnetic field is increased instead of developing into two separate gaps. When the twist angle deviates even further the correlation gap at $\nu = 3$ disappears (red data points in Fig. 3a; see Supplementary Fig. 5).

To reveal the effect of hetero-strain, we directly compare two regions with almost identical twist angle (-1.26°) but different hetero-strain values (0.10% versus 0.24%) for the same device, thus allowing other variables such as carrier concentration, electric field, and correlation strength to be kept mostly constant. In the region with a smaller hetero-strain (Fig. 3e), the $\nu = 3$ insulating gap develops into two branches under application of an out-of-plane magnetic field (Fig. 3f), indicating gate-induced switching between $C_{tot} = +2$ and $C_{tot} = -2$ QAH insulating states. In contrast, the region with a larger hetero-strain (Fig. 3g) exhibits only one branch of the $\nu = 3$ insulating gap (Fig. 3h; see Supplementary Fig. 6), corresponding to $C_{tot} = +2$ with no gate-controlled switching.

## Discussion

The emergence of correlated QAH insulating states in electron-doped tMBLG can be understood as resulting from spontaneous symmetry breaking driven by electron–electron Coulomb interactions. The CFB of tMBLG is fourfold degenerate due to spin and valley degrees of freedom. Each CFB sub-band in the unfolded graphene K+ (K−) valley hosts a non-zero Chern number of $C = +2$ (−2)

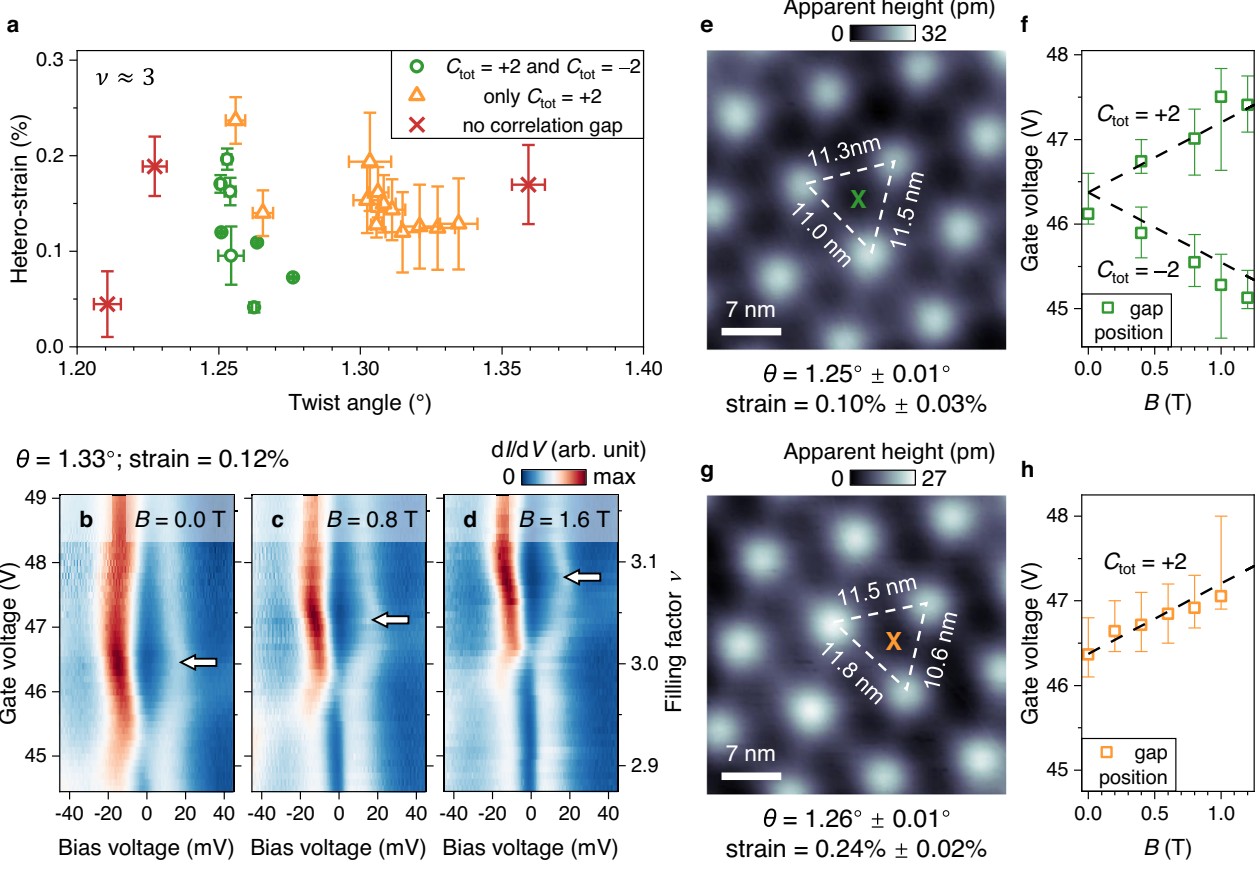

**Fig. 3 | Local structural effects on correlation and topology. a** Electronic phase diagram of tMBLG at $\nu = 3$ in the parameter space of local twist angle and local hetero-strain. $C_{tot}$ is the total Chern number. **b–d** Gate-dependent d$I$/d$V$ density plot for the BAB region at **b** $B = 0.0$ T, **c** $B = 0.8$ T, and **d** $B = 1.6$ T where only the $C_{tot} = +2$ gap is observed (modulation voltage $V_{RMS} = 1$ mV; setpoint $V_{Bias} = -75$ mV, $I_0 = 0.2$ nA). Arrows indicate correlation gaps. **e** STM topographic image of a region with $\theta = 1.25°$ and small hetero-strain of 0.10% ($V_{Bias} = -1$ V, $I_0 = 0.02$ nA). Standard deviations for angle and strain are calculated from uncertainties in the moiré wavelength. **f** Evolution of the correlation gap position near $\nu = 3$ in an out-of-plane

magnetic field shows two branches for small strain. The data points were extracted from d$I$/d$V$ spectra taken in the BAB region marked in (**e**) with the error bars determined via linear fitting (Supplementary Note 5). Dashed lines are guides to the eye following the Středa formula with $C_{tot} = \pm 2$. **g** STM topographic image of a region with $\theta = 1.26°$ and large hetero-strain of 0.24% ($V_{Bias} = -1$ V, $I_0 = 0.02$ nA). **h** Evolution of the correlation gap position near $\nu = 3$ in an out-of-plane magnetic field shows only one branch for large strain. The data points were extracted from d$I$/d$V$ spectra taken in the BAB region marked in (**g**). The dashed line is a guide to the eye following the Středa formula with $C_{tot} = +2$.

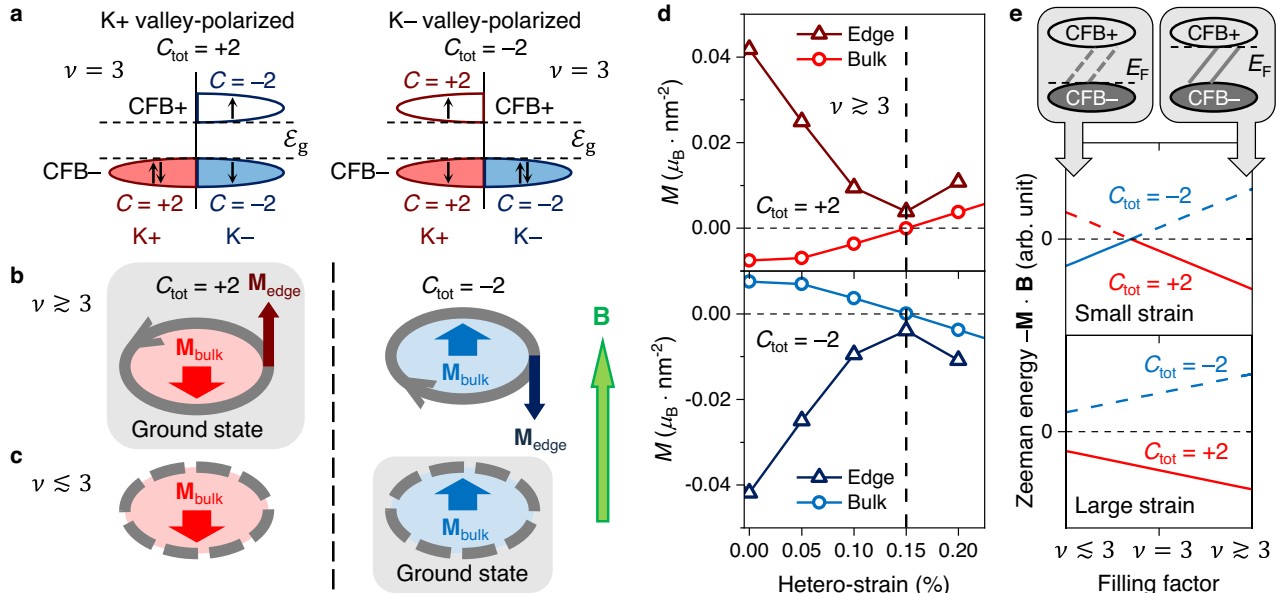

**Fig. 4 | Bulk–edge magnetization competition and gate-controlled Chern number switching. a** Energy configuration of the K+ valley-polarized state and the K− valley-polarized state at $\nu = 3$ in tMBLG. Arrows represent electron spin. $\mathscr{E}_g$ is the size of the correlation gap. The total Chern number $C_{tot}$ is the sum of Chern number $C$ for all occupied sub-bands. **b** Schematic of orbital magnetization for both valley-polarized states when $E_F$ is at the top of the correlation gap and the chiral edge state band is occupied ($\nu \gtrsim 3$). Grey arrows represent the direction of current flow in edge states. $M_{bulk}$ and $M_{edge}$ are bulk and edge orbital magnetic moments. Under an applied out-of-plane magnetic field **B** (green arrow on right) the $C_{tot} = +2$ state is the energetically favourable ground state. **c** Same as **b**, but $E_F$ is at the bottom of the correlation gap and the edge state band is depleted ($\nu \lesssim 3$). Dashed grey lines represent empty edge states. Here the $C_{tot} = -2$ state is the ground state. **d** Bulk and edge orbital magnetization for both valley-polarized states ($C_{tot} = \pm 2$) calculated based on a continuum model of 1.26° tMBLG. The vertical dashed line divides the parameter space into a small-strain regime where $M_{bulk}$ and $M_{edge}$ have opposite sign and a large-strain regime where they have the same sign. **e** Top sketch shows the depletion/occupation of the edge states (represented by dashed/solid grey lines) for different filling levels. Orbital Zeeman energy is plotted for both valley-polarized states ($C_{tot} = \pm 2$) as a function of filling factor when $E_F$ is inside the correlation gap. Solid (dashed) lines represent the energetically favourable (unfavourable) states. CFB− lower branch of conduction flat band, CFB+ upper branch of conduction flat band.

due to the large Berry curvature inherited from constituent graphene layers[18,19]. At integer fillings (e.g., $\nu = 2$ and $\nu = 3$) strong correlation can drive spontaneous polarization along a certain axis in the spin–valley space, splitting the CFB into occupied lower sub-bands (CFB−) and unoccupied upper sub-bands (CFB+) separated by a charge gap as observed in the experimental d$I$/d$V$ (Fig. 1e, g). At $\nu = 3$ the spin and valley polarization leads to breaking of time-reversal symmetry and topologically non-trivial states with $C_{tot} \neq 0$. Figure 4a shows one possible filling configuration with double occupancy of CFB sub-bands in the K+ valley, resulting in a QAH insulating state with $C_{tot} = +2$. Similarly, double occupancy of K− valley sub-bands can result in a $C_{tot} = -2$ state that is energetically equivalent to the $C_{tot} = +2$ state in the absence of an external magnetic field.

The large Berry curvature in the tMBLG moiré flat bands produces out-of-plane orbital magnetic moments that respond to an external magnetic field. The competition between bulk orbital magnetization ($M_{bulk}$) due to self-rotation of electron wave packets and edge orbital magnetization ($M_{edge}$) due to circulation of electrons in the topologically-protected in-gap chiral edge states is responsible for the gate-controlled switching behaviour at $\nu = 3$ (spin magnetism plays no explicit role in the switching due to negligible spin–orbit coupling in tMBLG)[8]. When the chemical potential resides in the bulk correlation gap the sign of $M_{edge}$ is the same as the sign of $C_{tot}$ and its magnitude is determined by how much the edge state band is filled[20]. The sign and magnitude of $M_{bulk}$, on the other hand, are sensitive to the detailed band structure and are not simply related to $C_{tot}$ other than the fact that a reversal in $C_{tot}$ is accompanied by a reversal in $M_{bulk}$ (Supplementary Note 6). The total orbital magnetization is the sum of $M_{bulk}$ and $M_{edge}$ and can be altered electrostatically by changing the filling of the edge states.

Our data is consistent with the special case illustrated in Fig. 4b, c which shows magnetic states for both $C_{tot} = +2$ and $C_{tot} = -2$ near $\nu = 3$ under the condition that $M_{bulk}$ and $M_{edge}$ are antiparallel and $|M_{edge}| > |M_{bulk}|$. When $E_F$ is placed at the top of the correlation gap ($\nu \gtrsim 3$, Fig. 4b), the chiral edge states are fully filled, and the total orbital magnetization is in the same direction as $M_{edge}$. When $E_F$ is moved to the bottom of the correlation gap ($\nu \lesssim 3$, Fig. 4c), however, the edge states are empty and the total magnetization is in the direction of $M_{bulk}$. In an external magnetic field the system will be driven into a magnetic ground state that aligns the total orbital magnetic moment with the applied field to minimize the orbital Zeeman energy. For the case shown here the energetically favourable states are $C_{tot} = +2$ for $\nu \gtrsim 3$ and $C_{tot} = -2$ for $\nu \lesssim 3$, thus illustrating how electrostatic gating can cause Chern number switching in tMBLG consistent with our experimental observations.

We have performed theoretical simulations that support the assumptions made in the switching picture described above (i.e., that $M_{bulk}$ and $M_{edge}$ are antiparallel and that $|M_{edge}| > |M_{bulk}|$) and that also explain why some tMBLG regions do not exhibit switching. For the experimental regimes shown in Fig. 3e–h (where the tMBLG twist angle lies close to 1.26°) we find that the key physical parameter that controls magnetic switching functionality is hetero-strain. To see this we calculated the strain-induced behaviour of $M_{bulk}$ and $M_{edge}$ for a continuum model of 1.26° tMBLG in the $\nu = 3$ QAH insulating state with $E_F$ set to the top of the correlation gap (a single phenomenological parameter characterizing the electron–electron interaction strength is included similar to ref. 8; see Supplementary Note 6 for details). Figure 4d shows a plot of the resulting $M_{edge}$ and $M_{bulk}$ for $C_{tot} = +2$ (top) and $C_{tot} = -2$ (bottom) as a function of hetero-strain with all other parameters kept constant. For small strain (<0.15%) $|M_{edge}|$ is significantly greater than $|M_{bulk}|$ while $M_{edge}$ and $M_{bulk}$ are anti-aligned. For

large hetero-strain (>0.15%), however, the situation changes significantly. The sign of $M_{edge}$ remains the same as $C_{tot}$, but $M_{bulk}$ flips its sign due to a strain-induced redistribution of Berry curvature and band dispersion throughout the mini-Brillouin zone. This divides the strain parameter space into two regimes separated by the vertical dashed line in Fig. 4d: the small-strain regime where $M_{bulk}$ and $M_{edge}$ have opposite sign (and switching can occur), and the large-strain regime where they have the same sign (and switching does not occur). To better illustrate this behaviour, Fig. 4e shows a schematic of the resulting orbital Zeeman energy for $B > 0$ as a function of filling factor near $\nu = 3$. Here the solid lines show the energetically favourable ground state and the dashed lines show the unfavourable state having opposite $C_{tot}$. For small hetero-strain $M_{bulk}$ dominates for $\nu \lesssim 3$ (when the edge state band is empty) and results in $C_{tot} = -2$ whereas $M_{edge}$ dominates for $\nu \gtrsim 3$ (as the edge states are filled), resulting in a switching to $C_{tot} = +2$. For large hetero-strain, $M_{bulk}$ and $M_{edge}$ both line up parallel to the applied field at all fillings and the system stays at $C_{tot} = +2$ without switching, consistent with our experimental observations.

In conclusion, we have observed local spectroscopic signatures of strong correlation and non-trivial topology at the moiré scale in tMBLG, and have demonstrated local control of the QAH Chern number via electrostatic gating. Combining STM and STS allows us to characterize the topological and magnetic switching phase diagram of tMBLG in the parameter space of local twist angle and hetero-strain. We observe magnetic switching only at low strain, revealing the sensitive interplay between correlation, topology, and local structural parameters that determines electronic and magnetic ground states in twisted moiré systems. This provides insight into the many phase diagrams observed in related moiré systems via measurements that average over regions having different structural parameters, and creates new opportunities for future manipulation of QAH insulating domains and the chiral edge states that lie between them.

## Methods

### Sample preparation
Samples were prepared using the "flip-chip" method[21] followed by a forming-gas anneal[22,23]. Electrical contacts were made by evaporating Cr/Au (5 nm/70 nm) through a silicon nitride shadow-mask onto the heterostructure. The sample surface cleanliness was confirmed using contact-AFM prior to STM measurements. Samples were annealed at 300 °C overnight in ultra-high vacuum before insertion into the low-temperature STM stage.

### STM/STS measurements
STM/STS measurements were performed in a commercial CreaTec LT-STM held at $T = 4.7$ K using tungsten (W) tips. STM tips were prepared on a Cu(111) surface and calibrated against the Cu(111) Shockley surface state before measurements to avoid tip artifacts. A voltage $V_G$ was applied to the Si back-gate to change the carrier density $n$ in the tMBLG stack ($n = \frac{\varepsilon_D \varepsilon_0 V_G}{e d_D}$ where $\varepsilon_D \approx 3.6$ is the average out-of-plane dielectric constant of hBN and SiO₂, $\varepsilon_0$ is the vacuum permittivity, $e$ is the elementary charge, and $d_D = 335$ nm is the thickness of the dielectric layers). d$I$/d$V$ spectra were recorded using standard lock-in techniques with a small bias modulation $V_{RMS} = 1$ mV at 613 Hz. All STM images were edited using WSxM software[24].

### Determination of local twist angle and hetero-strain
The local twist angle and hetero-strain were determined by analysing the monolayer–bilayer moiré pattern. The moiré wavelength was obtained for three directions by measuring the spatial separation between peaks in STM topographs and averaging over several moiré unit cells. The reciprocal primitive vectors $K_i$ ($i = 0, 1, 2$) were derived through Fourier transform analysis. In the presence of hetero-strain, $K_i$ can be approximately written as $K_i = k\left(\theta + (1 + \nu_P)\epsilon \cos\left(\alpha + i\frac{2\pi}{3}\right) \sin\left(\alpha + i\frac{2\pi}{3}\right)\right)$ where $k = 4.694$ nm$^{-1}$ is the

length of the graphene reciprocal primitive vectors, $\theta$ is the twist angle, $\epsilon$ is the hetero-strain amplitude, $\nu_P = 0.16$ is Poisson's ratio for graphene, and $\alpha$ is the angle between the principal axis of the strain tensor and one of the graphene reciprocal primitive vectors[25]. This allows us to solve both $\theta$ and $\epsilon$ from the extracted $K_i$.

## Data availability
The data that support the plots within this paper and the findings of this study are provided in the Source Data file. Source data are provided with this paper.

## Code availability
The computer codes that support the plots within this paper and the findings of this study are available from the corresponding authors upon request.

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

## Acknowledgements

The authors thank Birui Yang for technical support. This research was supported by the Center for Novel Pathways to Quantum Coherence in Materials, an Energy Frontier Research Center funded by the US Department of Energy, Office of Science, Basic Energy Sciences (STM/STS measurements and analysis). Support was also provided by the Director, Office of Science, Office of Basic Energy Sciences, Materials Sciences and Engineering Division of the US Department of Energy under contract number DE-AC02-05CH11231 within the van der Waals Heterostructures program KCWF16 (device architecture development); by the Molecular Foundry at LBNL, which is funded by the Director, Office of Science, Office of Basic Energy Sciences, Scientific User Facilities Division, of the US Department of Energy under Contract No. DE-AC02-05CH11231 (graphene layer characterization); by the National Science Foundation Award DMR-2221750 (device AFM characterization); and by the US Department of Energy, Office of Science, Office of Basic Energy Sciences, Materials Sciences and Engineering Division under Contract No. DE-AC02-05-CH11231 within the Theory of Materials program KC2301 (tMBLG simulations). K.W. and T.T. acknowledge support from JSPS KAKENHI (Grant Numbers 20H00354, 21H05233, and 23H02052) and World Premier International Research Center Initiative (WPI), MEXT, Japan (hBN crystal synthesis and characterization). C.Z. acknowledges support from a Kavli ENSI Philomathia Graduate Student Fellowship. T.S. acknowledges fellowship support from the Masason Foundation.

## Author contributions

C.Z., T.Z., S.K., and M.F.C. initiated and conceived the research. C.Z. and T.Z. carried out STM/STS measurements and analyses. M.F.C. supervised STM/STS measurements and analyses. S.K. prepared gate-tunable devices. A.Z., F.W., and M.F.C. supervised device preparations. T.T. and K.W. provided the hBN crystals. T.S. performed theoretical calculations and analyses. M.P.Z. supervised theoretical calculations and analyses. C.Z., T.Z., and M.F.C. wrote the manuscript with help from all authors. All authors contributed to the scientific discussion.

## Competing interests

The authors declare no competing interests.
