## [Peer review file · Nature Communications]

REVIEWER COMMENTS

Reviewer #1 (Remarks to the Author):

In the present manuscript, "Local spectroscopy of a gate-switchable Moir'\{e} quantum anomalous Hall insulator," authors have investigated an exciting problem in a graphene superlattice system.

By use of scanning tunneling microscopy and spectroscopy, authors have demonstrated how the interplay

between correlation, topology, and local atomic structure determines the behavior of electron-doped twisted monolayer-bilayer graphene (tMBLG). Namely, in the gate- and magnetic-field-dependent measurements, the QAH insulating state having total Chern number $C_{\text{tot}} = \pm 2$ at a doping level of three electrons per Moir'\{e} unit cell has been observed. Moreover, using strains, the transition between phases with different Chern numbers was demonstrated, indicating the dependence of phase structure on the local parameters of the physical system.

The paper is sound for publication in Nature Communications. It will stimulate theoretical studies trying to revile the response of the system to various types of deformation not only of the tMBLG type superlattices but also in monolayer graphene itself. Hence the paper is interesting for a broad community of condensed matter physicists. The paper is well written. The presence of intuitive arguments and explanations for why strains/deformations lead to a topology change would be helpful.

Reviewer #2 (Remarks to the Author):

The authors report scanning tunneling spectroscopy measurements of twisted monolayer-bilayer graphene as a function of doping and magnetic field. Their results reveal the interplay between topological behavior and the local strain and twist angle in the sample. This is something that is difficult to address without the local probe results provided by scanning tunneling microscopy. The results are intriguing but not completely convincing. In particular, the model that strain suppresses the switching behavior does not have much experimental support.

It seems that twist angle is more important for the gate tunability of the quantum anomalous Hall states than the local strain. For twist angles between 1.25 and 1.28 degrees, two values of the Chern number are observed for all but two values of the strain. While outside this regime, this never occurs even for small strain. The manuscript focuses on one value of strain, 0.24% where the Chern number is not tunable. However, there is also a value around 0.15% that is not tunable while values around 0.20% are tunable. So, clearly there is sample to sample variation and it is not just high strain that is not tunable. So how can strong conclusions be drawn with the limited number of samples that do not show tunability in this twist angle regime?

Are there calculations like in Fig. 4d and 4e that show that the Chern number is always +2 for all strain values when the twist angle is away from 1.25 degrees? This would be good supporting evidence that it is a strain effect and not a twist angle effect.

Why are no changes in dI/dV spectroscopy at negative filling factors (except $\nu=-4$) observed?

Also, there is no feature present at $\nu=+1$ where the bands should be partially filled. This is explained by the fact that the charge density and electric field can not be tuned independently with only a single gate electrode. Can the height of the tip (tunnel current) not be used to control the electrostatic environment and hence the electric field at a give charge density? This should allow the parameters to be tuned such that the $\nu=1$ state is visible.

Do you observe any hysteresis in the dI/dV signal as the magnetic field is changed? Do changes in the signal always occur at the same filling factor regardless of direction of the magnetic field sweep?

Reviewer #3 (Remarks to the Author):

The authors used STM to study the interplay between correlation, topology, and local atomic structure on the electronic properties of tMBLG. The study in this work reveals the relationship between topological behavior and local twist angle accompanied by hetero-strain. Such results are helpful to understand inconsistent phase diagrams reported in tMBLG devices via macroscopic probes and are

quite interesting and deserve to be published in Nature Communications. However, there are three issues that need to be addressed.

1. The authors should compare the QAH phase in the tMBLG with that in magic-angle TBG in details. The differences between them should be discussed and what is the origin of their differences?

2. Because of existence of hetero-strain, the electronic properties in the tMBLG are quite different. Is there any experimental evidence of the QAH domain and the chiral edge states along the edge of the domain?

3. In previous study (PRB 102, 121406(R) 2020), an increase of the insulating gap in response to the magnetic field is observed in partially filled flat bands and such a behavior is attributed to the orbital magnetic moments in each moire. Is there any similar feature in the tMBLG? Since the results in Figs. 2 and 3 are arising from the orbital magnetic moments. I believe a detailed discussion would be helpful for the readers.

Reviewer #1 (Remarks to the Author):

In the present manuscript, "Local spectroscopy of a gate-switchable Moiré quantum anomalous Hall insulator", authors have investigated an exciting problem in a graphene superlattice system. By use of scanning tunneling microscopy and spectroscopy, authors have demonstrated how the interplay between correlation, topology, and local atomic structure determines the behavior of electron-doped twisted monolayer-bilayer graphene (tMBLG). Namely, in the gate- and magnetic-field-dependent measurements, the QAH insulating state having total Chern number $C_{\text{tot}} = \pm 2$ at a doping level of three electrons per Moiré unit cell has been observed. Moreover, using strains, the transition between phases with different Chern numbers was demonstrated, indicating the dependence of phase structure on the local parameters of the physical system.

The paper is sound for publication in Nature Communications. It will stimulate theoretical studies trying to revile the response of the system to various types of deformation not only of the tMBLG type superlattices but also in monolayer graphene itself. Hence the paper is interesting for a broad community of condensed matter physicists. The paper is well written. The presence of intuitive arguments and explanations for why strains/deformations lead to a topology change would be helpful.

Response: We thank the reviewer for the positive comments. We have added an intuitive explanation for strain effects on topological behavior to the manuscript. The gate-controlled switching of topological states arises from a competition between M_{bulk} and M_{edge} , and can only occur when M_{bulk} and M_{edge} are antiparallel with $|M_{\text{edge}}| > |M_{\text{bulk}}|$. Increasing the tMBLG hetero-strain causes a redistribution of Berry curvature and band dispersion throughout the moiré Brillouin zone. As a result, M_{bulk} is strongly modified and eventually becomes parallel to M_{edge} when the hetero-strain is large enough, thus suppressing the topological switching. To emphasized this point, we have included the above argument in the main text as " M_{bulk} flips its sign due to a strain-induced redistribution of Berry curvature and band dispersion throughout the moiré Brillouin zone" (Lines 211-213) and have also included more detailed explanations in the Method section (Page 18).

Reviewer #2 (Remarks to the Author):

The authors report scanning tunneling spectroscopy measurements of twisted monolayer-bilayer graphene as a function of doping and magnetic field. Their results reveal the interplay between topological behavior and the local strain and twist angle in the sample. This is something that is difficult to address without the local probe results provided by scanning tunneling microscopy. The results are intriguing but not completely convincing. In particular, the model that strain suppresses the switching behavior does not have much experimental support.

It seems that twist angle is more important for the gate tunability of the quantum anomalous Hall states than the local strain. For twist angles between 1.25 and 1.28 degrees, two values of the Chern number are observed for all but two values of the strain. While outside this

regime, this never occurs even for small strain. The manuscript focuses on one value of strain, 0.24% where the Chern number is not tunable. However, there is also a value around 0.15% that is not tunable while values around 0.20% are tunable. So, clearly there is sample to sample variation and it is not just high strain that is not tunable. So how can strong conclusions be drawn with the limited number of samples that do not show tunability in this twist angle regime?

Response: We agree that twist angle is one of the important parameters controlling the correlated and topological behavior in moiré systems, as demonstrated via transport and optical measurements in several published results that we have referenced. The data presented in this manuscript also shows the significant impact of local twist angle on gate-controlled topological switching of tMBLG. However, other parameters such as hetero-strain, dielectric screening, and out-of-plane electric field can also affect the electronic properties of moiré systems. In most cases these parameters cannot be independently varied, making it difficult to determine their individual impact.

In this manuscript, we present a *controlled experiment* focused on how hetero-strain shapes the interplay between correlation and topology in tMBLG. This effect has not been systematically studied before because, as the reviewer mentioned, it requires the unique ability of STM/STS to simultaneously characterize local structural and electronic properties. We have carefully chosen two tMBLG regions that have almost identical twist angle ($\sim 1.26^\circ$) and that are on the same device only $\sim 1 \mu\text{m}$ apart, with the only difference between them being the hetero-strain (0.10% vs. 0.24%) (Fig. 3e-h). This ensures that all other relevant parameters such as twist angle, screening environment, and electric field are kept nearly constant, enabling the strain effects to be isolated. We have also theoretically analyzed the impact of hetero-strain on the orbital magnetization of tMBLG QAH states, thus providing a consistent explanation for our experimental data.

We agree with the reviewer that “it is not just high strain” that matters when considering other data points in our measurements. Our manuscript is not intended to answer the question “which parameter has a larger impact on the switching behavior”, and we are not claiming that hetero-strain plays the most important role. The data points the reviewer mentioned (the value around 0.15% that is not switchable and values around 0.20% that are switchable) come from two different samples that have different twist angle values ($\sim 1.25^\circ$ vs. $\sim 1.27^\circ$), so their contrasting behavior arises from a combination of the difference in twist angle, hetero-strain, and screening environment. Our focus has been to isolate the effect of strain. A full understanding of the individual effect of all other parameters would require additional controlled experiments, such as actively changing the twist angle on the same device while maintaining a constant strain. This exceeds the scope of this work and is also beyond the state-of-the-art of STM/STS.

We have made changes to the manuscript to clarify these points. We have added the sentence “in addition to observing strong variation of correlated and topological properties at different twist angles, we find that regions having nearly identical twist angle but different hetero-strain values exhibit very different behavior” in the introductory paragraph (Lines 62-65), and also added “with all other parameters kept constant” in the “Theoretical model and discussion” section (Line 209).

Are there calculations like in Fig. 4d and 4e that show that the Chern number is always +2 for all strain values when the twist angle is away from 1.25 degrees? This would be good supporting evidence that it is a strain effect and not a twist angle effect.

Response: We agree that twist angle is an important parameter and are not claiming that hetero-strain is the only factor shaping the gate-controlled topological switching behavior. Our goal is to show how hetero-strain can affect the QAH states of tMBLG when twist angle and other relevant parameters remain the same. We focus on $\theta = 1.26^\circ$ in the manuscript because our experimental data for that twist angle clearly shows switchable states at low strain and non-switchable states at high strain. This is further supported by our theoretical analysis, thus revealing the *isolated* effect of hetero-strain in controlling the topological switching of tMBLG.

Although we have not observed switching behavior for twist angles away from 1.26° , we find it difficult to draw strong conclusions from those data points. Experimentally we did not find low-strain regions ($< 0.1\%$) within the twist angle range $1.30^\circ < \theta < 1.34^\circ$.

As requested by the reviewer we have calculated M_{bulk} for different twist angle and hetero-strain values (the sign of M_{bulk} directly determines whether switching can happen). In Fig. R1 below we show the strain dependence of $M_{\text{bulk}}(C_{\text{tot}} = +2)$ at $\theta = 1.26^\circ, 1.28^\circ$, and 1.32° for different choices of δ_{VFB} (δ_{VFB} is related to the electron-electron interaction strength as defined in Methods (Lines 378-379)). For each twist angle, increasing the hetero-strain makes $M_{\text{bulk}}(C_{\text{tot}} = +2)$ significantly larger due to a redistribution of Berry curvature and band dispersion throughout the moiré Brillouin zone. The transition from $M_{\text{bulk}}(C_{\text{tot}} = +2) < 0$ to $M_{\text{bulk}}(C_{\text{tot}} = +2) > 0$ (i.e., from switchable to non-switchable), therefore, might occur *if* δ_{VFB} is within a certain range. Our current theoretical capabilities, however, do not allow us to predict whether this transition *actually* occurs at arbitrary twist angle due to uncertainty in δ_{VFB} which involves many-body effects and can vary strongly with twist angle. We have added this discussion to the Methods section (Page 18) and included Fig. R1 as the new Extended Data Fig. 9.

Figure R1: Bulk magnetization for different twist angle and hetero-strain values. a, Energy configuration of the K+ valley-polarized state at $\nu = 3$ showing sub-bands from both the CFB manifold and the VFB manifold. δ_{VFB} is the interaction-induced energy offset among the VFB sub-bands. **b-d**, M_{bulk} for $C_{\text{tot}} = +2$ state as a function of hetero-strain for different choices of δ_{VFB} at **(b)** $\theta = 1.26^\circ$, **(c)** 1.28° , and **(d)** 1.32° .

Why are no changes in dI/dV spectroscopy at negative filling factors (except $\nu = -4$) observed?

Response: We did not observe correlated states on the hole-doping side. This is consistent with transport studies on tMBLG (see, for example, *Nature* **588**, 66 (2020)) and is likely because the valence flat band is not as flat as the conduction flat band.

Also, there is no feature present at $\nu = +1$ where the bands should be partially filled. This is explained by the fact that the charge density and electric field can not be tuned independently with only a single gate electrode. Can the height of the tip (tunnel current) not be used to control the electrostatic environment and hence the electric field at a give charge density? This should allow the parameters to be tuned such that the $\nu = 1$ state is visible.

Response: We agree that tip-induced gating can in principle be utilized to tune the electric field independent of the charge carrier density. However, this would require a tip-sample work function mismatch, which would make it very hard to detect the tMBLG correlated states via STS in a straightforward way. In our experiments we have deliberately used a tungsten (W) tip and prepared it on a Cu(111) surface to match the tip/surface work functions (since Cu has a work function similar to graphene (~ 4.7 eV)). This allows us to minimize tip-induced gating – an essential condition for observing correlation gaps via STS, but unfortunately prohibits us from exploring the electric field effect by controlling the tip height. To clarify this we have added a sentence “tip-induced gating is not observed in our experiment due to deliberate work function matching between graphene and the tip material” to the Methods section (Lines 262-263).

Do you observe any hysteresis in the dI/dV signal as the magnetic field is changed? Do changes in the signal always occur at the same filling factor regardless of direction of the magnetic field sweep?

Response: We do not observe any hysteretic behavior within our detection limit (as shown in Fig. 3f,h). The correlation gaps appear at particular filling factors according to the Středa formula $\frac{\Delta n}{\Delta B} = \frac{C_{\text{tot}}}{\Phi_0}$. This is a topological/adiabatic (not dynamic) effect, so it should not depend on the direction of B -field sweep or exhibit hysteresis behavior.

Reviewer #3 (Remarks to the Author):

The authors used STM to study the interplay between correlation, topology, and local atomic structure on the electronic properties of tMBLG. The study in this work reveals the relationship between topological behavior and local twist angle accompanied by hetero-strain. Such results are helpful to understand inconsistent phase diagrams reported in tMBLG devices via macroscopic probes and are quite interesting and deserve to be published in Nature Communications. However, there are three issues that need to be addressed.

1. The authors should compare the QAH phase in the tMBLG with that in magic-angle TBG in details. The differences between them should be discussed and what is the origin of their differences?

Response: The QAH phase in magic-angle twisted bilayer graphene (MA-tBLG) has only been observed in samples aligned with the hBN substrate (*Science* **367**, 900 (2020)). The underlying physics of the QAH phase in tMBLG and that in hBN-aligned MA-tBLG are similar, with two major differences as reported in transport experiments. First, the Hall conductance is quantized to $\pm e^2/h$ for tBLG and $\pm 2e^2/h$ for tMBLG, since the total Chern number for QAH states of tBLG is ± 1 while that for tMBLG is ± 2 . Second, gate-induced Chern number switching has only been observed in tMBLG. A similar gate-induced anomalous Hall switching has been observed in MA-tBLG, but at a filling level where no quantized Hall conductance is observed. This likely depends on details of the band structure, as discussed in a previous theoretical work (*PRL* **125**, 227702 (2020)). In terms of local spectroscopy, Chern insulating states have been observed in tBLG under high magnetic fields, but no systematic STM/STS study has been

performed on the QAH phase of hBN-aligned MA-tBLG yet. We have added a Supplementary Note titled “Comparison between QAH phases in tMBLG and MA-tBLG” where we address these points.

2. Because of existence of hetero-strain, the electronic properties in the tMBLG are quite different. Is there any experimental evidence of the QAH domain and the chiral edge states along the edge of the domain?

Response: The dependence of topological behavior on hetero-strain observed in this manuscript indeed opens new possibilities for realizing QAH domains and chiral edge states, but pursuing those effects is an independent study that goes beyond the scope of this manuscript. In a follow-up study (arXiv:2212.03380) we have shown that edge-state imaging can be achieved with a fixed hetero-strain through the use of spatially-inhomogeneous carrier density distributions.

3. In previous study (PRB 102, 121406(R) 2020), an increase of the insulating gap in response to the magnetic field is observed in partially filled flat bands and such a behavior is attributed to the orbital magnetic moments in each moiré. Is there any similar feature in the tMBLG? Since the results in Figs. 2 and 3 are arising from the orbital magnetic moments. I believe a detailed discussion would be helpful for the readers.

Response: We agree that such an orbital Zeeman effect could change the insulating gap size as the B -field is increased. On the other hand, unlike the case in *PRB* **102**, 121406(R) (2020) where each Chern sector is either completely occupied or completely empty, for the $\nu = 3$ correlated insulating states in tMBLG one Chern sector is doubly (i.e., fully) occupied whereas the other is singly (i.e., half) occupied. Whether and how much the gap size changes therefore depends on multiple factors such as valley polarization and orbital magnetization of each sub-band. The relevant orbital magnetic moment per moiré unit cell is usually less than 8 Bohr magnetons as shown in our calculations (Fig. 4, Extended Data Fig. 9), so the corresponding gap size change could not exceed 2 meV at $B = 2$ T (the maximum B -field that can be achieved for our instrument). This is roughly the same as thermal and instrumental broadening, consistent with the fact that we did not observe any significant difference in gap size at different B -fields. We have added a Supplementary Note titled “Response of the $\nu = 3$ correlation gap size to an applied out-of-plane B -field” to discuss this issue.

REVIEWERS' COMMENTS

Reviewer #2 (Remarks to the Author):

The authors have provided satisfactory explanations for the questions from my previous review as well as provided further calculations. The manuscript is suitable for Nature Communications.

Reviewer #3 (Remarks to the Author):

The authors have addressed my criticisms and have answered my questions. I now recommend this manuscript for publication in Nature Communications.